# STATE SPACE LSTM MODELS
## WITH PARTICLE MCMC INFERENCE

## ABSTRACT

Long Short-Term Memory (LSTM) is one of the most powerful sequence models. Despite the strong performance, however, it lacks the nice interpretability as in state space models. In this paper, we present a way to combine the best of both worlds by introducing State Space LSTM (SSL) models that generalizes the earlier work Zaheer et al. (2017) of combining topic models with LSTM. However, unlike Zaheer et al. (2017), we do not make any factorization assumptions in our inference algorithm. We present an efficient sampler based on sequential Monte Carlo (SMC) method that draws from the joint posterior directly. Experimental results confirms the superiority and stability of this SMC inference algorithm on a variety of domains.

## 1 INTRODUCTION

State space models (SSMs), such as hidden Markov models (HMM) and linear dynamical systems (LDS), have been the workhorse of sequence modeling in the past decades From a graphical model perspective, efficient message passing algorithms (Stratonovich, 1960; Kalman, 1960) are available in compact closed form thanks to their simple linear Markov structure. However, simplicity comes at a cost: real world sequences can have long-range dependencies that cannot be captured by Markov models; and the linearity of transition and emission restricts the flexibility of the model for complex sequences.

A popular alternative is the recurrent neural networks (RNN), for instance the Long Short-Term Memory (LSTM) (Hochreiter & Schmidhuber, 1997) which has become a standard for sequence modeling nowadays. Instead of associating the observations with stochastic latent variables, RNN directly defines the distribution of each observation conditioned on the past, parameterized by a neural network. The recurrent parameterization not only allows RNN to provide a rich function class, but also permits scalable stochastic optimization such as the backpropagation through time (BPTT) algorithm. However, flexibility does not come for free as well: due to the complex form of the transition function, the hidden states of RNN are often hard to interpret. Moreover, it can require large amount of parameters for seemingly simple sequence models (Zaheer et al., 2017).

In this paper, we propose a new class of models *State Space LSTM (SSL)* that combines the best of both worlds. We show that SSLs can handle nonlinear, non-Markovian dynamics like RNNs, while retaining the probabilistic interpretations of SSMs. The intuition, in short, is to separate the state space from the sample space. In particular, instead of directly estimating the dynamics from the observed sequence, we focus on modeling the sequence of latent states, which may represent the true underlying dynamics that generated the noisy observations. Unlike SSMs, where the same goal is pursued under linearity and Markov assumption, we alleviate the restriction by directly modeling the transition function between states parameterized by a neural network. On the other hand, we bridge the state space and the sample space using classical probabilistic relation, which not only brings additional interpretability, but also enables the LSTM to work with more structured representation rather than the noisy observations.

Indeed, parameter estimation of such models can be nontrivial. Since the LSTM is defined over a sequence of latent variables rather than observations, it is not straightforward to apply the usual BPTT algorithm without making variational approximations. In Zaheer et al. (2017), which is an instance of SSL, an EM-type approach was employed: the algorithm alternates between imputing the latent

---
*Work done while intern at Google.

states and optimizing the LSTM over the imputed sequences. However, as we show below, the inference implicitly assumes the posterior is factorizable through time. This is a restrictive assumption since the benefit of rich state transition brought by the LSTM may be neutralized by breaking down the posterior over time.

We present a general parameter estimation scheme for the proposed class of models based on sequential Monte Carlo (SMC) (Doucet et al., 2001), in particular the Particle Gibbs (Andrieu et al., 2010). Instead of sampling each time point individually, we directly sample from the joint posterior without making limiting factorization assumptions. Through extensive experiments we verify that sampling from the full posterior leads to significant improvement in the performance.

**Related works**   Enhancing state space models using neural networks is not a new idea. Traditional approaches can be traced back to nonlinear extensions of linear dynamical systems, such as extended or unscented Kalman filters (Julier & Uhlmann, 1997), where both state transition and emission are generalized to nonlinear functions. The idea of parameterizing them with neural networks can be found in Ghahramani & Roweis (1999), as well as many recent works (Krishnan et al., 2015; Archer et al., 2015; Johnson et al., 2016; Krishnan et al., 2017; Karl et al., 2017) thanks to the development of recognition networks (Kingma & Welling, 2014; Rezende et al., 2014). Enriching the output distribution of RNN has also regain popularity recently. Unlike conventionally used multinomial output or mixture density networks (Bishop, 1994), recent approaches seek for more flexible family of distributions such as restricted Boltzmann machines (RBM) (Boulanger-Lewandowski et al., 2012) or variational auto-encoders (VAE) (Gregor et al., 2015; Chung et al., 2015).

On the flip side, there have been studies in introducing stochasticity to recurrent neural networks. For instance, Pachitariu & Sahani (2012) and Bayer & Osendorfer (2014) incorporated independent latent variables at each time step; while in Fraccaro et al. (2016) the RNN is attached to both latent states and observations. We note that in our approach the transition and emission are decoupled, not only for interpretability but also for efficient inference without variational assumptions.

On a related note, sequential Monte Carlo methods have recently received attention in approximating the variational objective (Maddison et al., 2017; Le et al., 2017; Naesseth et al., 2017). Despite the similarity, we emphasize that the context is different: we take a stochastic EM approach, where the full expectation in E-step is replaced by the samples from SMC. In contrast, SMC in above works is aimed at providing a tighter lower bound for the variational objective.

## 2   BACKGROUND

In this section, we provide a brief review of some key ingredients of this paper. We first describe the SSMs and the RNNs for sequence modeling, and then outline the SMC methods for sampling from a series of distributions.

### 2.1   STATE SPACE MODELS

Consider a sequence of observations $x_{1:T} = (x_1, \ldots, x_T)$ and a corresponding sequence of latent states $z_{1:T} = (z_1, \ldots, z_T)$. The SSMs are a class of graphical models that defines probabilistic dependencies between latent states and the observations. A classical example of SSM is the (Gaussian) LDS, where real-valued states evolve linearly over time under the first-order Markov assumption. Let $x_t \in \mathbb{R}^d$ and $z_t \in \mathbb{R}^k$, the LDS can be expressed by two equations:

$$\text{(Transition)} \quad z_t = A z_{t-1} + \eta, \quad \eta \sim \mathcal{N}(0, Q) \tag{1}$$

$$\text{(Emission)} \quad x_t = C z_t + \epsilon, \quad \epsilon \sim \mathcal{N}(0, R), \tag{2}$$

where $A \in \mathbb{R}^{k \times k}$, $C \in \mathbb{R}^{d \times k}$, and $Q$ and $R$ are covariance matrices of corresponding sizes. They are widely applied in modeling the dynamics of moving objects, with $z_t$ representing the true state of the system, such as location and velocity of the object, and $x_t$ being the noisy observation under zero-mean Gaussian noise.

We mention two important inference tasks (Koller & Friedman, 2009) associated with SSMs. The first tasks is *filtering*: at any time $t$, compute $p(z_t|x_{1:t})$, *i.e.* the most up-to-date belief of the state $z_t$ conditioned on all past and current observations $x_{1:t}$. The other task is *smoothing*, which computes $p(z_t|x_{1:T})$, *i.e.* the update to the belief of a latent state by incorporating future observations. One of the beauties of SSMs is that these inference tasks are available in closed form, thanks

---

**Algorithm 1** Sequential Monte Carlo

    1. Let $z_0^p = z_0$ and weights $\alpha_0^p = 1/P$ for $p = 1, \ldots, P$.

    2. For $t = 1, \ldots, T$:

        (a) Sample ancestors $a_{t-1}^p \sim \alpha_{t-1}$ for $p = 1, \ldots, P$.

        (b) Sample particles $z_t^p \sim f(z_t | z_{1:t-1}^{a_{t-1}^p})$ for $p = 1, \ldots, P$.

        (c) Set $z_{1:t}^p = (z_{1:t-1}^{a_{t-1}^p}, z_t^p)$ for $p = 1, \ldots, P$.

        (d) Compute the normalized weights $\alpha_t^p \propto \dfrac{\pi_t(z_{1:t}^p)}{\pi_{t-1}(z_{1:t-1}^{a_{t-1}^p}) f(z_t^p | z_{1:t-1}^{a_{t-1}^p})}$ for $p = 1, \ldots, P$.

---

to the simple Markovian dynamics of the latent states. For instance, the forward-backward algorithm (Stratonovich, 1960), the Kalman filter (Kalman, 1960), and RTS smoother (Rauch et al., 1965) are widely appreciated in the literature of HMM and LDS.

Having obtained the closed form filtering and smoothing equations, one can make use of the EM algorithm to find the maximum likelihood estimate (MLE) of the parameters given observations. In the case of LDS, the E-step can be computed by RTS smoother and the M-step is simple subproblems such as least-squares regression. We refer to Ghahramani & Hinton (1996) for a full exposition on learning the parameters of LDS using EM iterations.

## 2.2 RECURRENT NEURAL NETWORKS

RNNs have received remarkable attention in recent years due to their strong benchmark performance as well as successful applications in real-world problems. Unlike SSMs, RNNs aim to directly learn the complex generative distribution of $p(x_t | x_{1:t-1})$ using a neural network, with the help of a deterministic internal state $s_t$:

$$p(x_t | x_{1:t-1}) = p(x_t; g(s_t)), \quad s_t = \mathsf{RNN}(s_{t-1}, x_{t-1}), \tag{3}$$

where $\mathsf{RNN}(\cdot, \cdot)$ is the transition function defined by a neural network, and $g(\cdot)$ is an arbitrary differentiable function that maps the RNN state $s_t$ to the parameter of the distribution of $x_t$. The flexibility of the transformation function allows the RNN to learn from complex nonlinear non-Gaussian sequences. Moreover, since the state $s_t$ is a deterministic function of the past observations $x_{1:t-1}$, RNNs can capture long-range dependencies, for instance matching brackets in programming languages (Karpathy et al., 2015).

The BPTT algorithm can be used to find the MLE of the parameters of $\mathsf{RNN}(\cdot, \cdot)$ and $g(\cdot)$. However, although RNNs can, in principle, model long-range dependencies, directly applying BPTT can be difficult in practice since the repeated application of a squashing nonlinear activation function, such as tanh or logistic sigmoid, results in an exponential decay in the error signal through time. LSTMs (Hochreiter & Schmidhuber, 1997) are designed to cope with the such vanishing gradient problems, by introducing an extra memory cell that is constructed as a linear combination of the previous state and signal from the input. In this work, we also use LSTMs as building blocks, as in Zaheer et al. (2017).

## 2.3 SEQUENTIAL MONTE CARLO

Sequential Monte Carlo (SMC) (Doucet et al., 2001) is an algorithm that samples from a series of potentially unnormalized densities $\pi_1(z_1), \ldots, \pi_T(z_{1:T})$. At each step $t$, SMC approximates the target density $\pi_t$ with $P$ weighted particles using importance distribution $f(z_t | z_{1:t-1})$:

$$\pi_t(z_{1:t}) \approx \hat{\pi}_t(z_{1:t}) = \sum_p \alpha_t^p \delta_{z_{1:t}^p}(z_{1:t}), \tag{4}$$

where $\alpha_t^p$ is the importance weight of the $p$-th particle and $\delta_x$ is the Dirac point mass at $x$. Repeating this approximation for every $t$ leads to the SMC method, outlined in Algorithm 1.

The key to this method lies in the resampling, which is implemented by repeatedly drawing the ancestors of particles at each step. Intuitively, it encourages the particles with a higher likelihood to

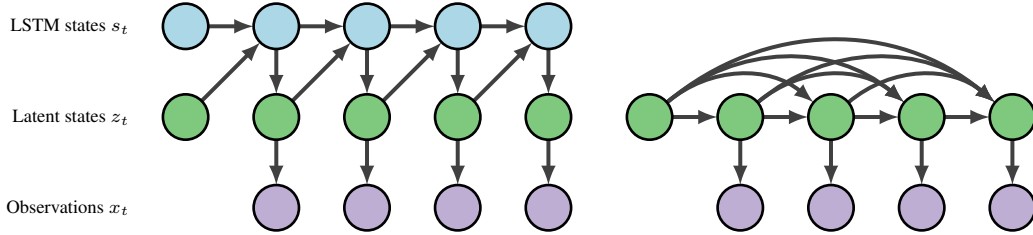

(a) Full model with explicit LSTM states.      (b) Model after collapsing LSTM states.

Figure 1: Generative process of SSL.

survive longer, since the weight reflects the likelihood of the particle path. The final Monte Carlo estimate (4) consists of only survived particle paths, and sampling from this point masses is equivalent to choosing a particle path according to the last weights $\alpha_T$. We refer to Doucet et al. (2001); Andrieu et al. (2010) for detailed proof of the method.

## 3 STATE SPACE LSTM MODELS

In this section, we present the class of State Space LSTM (SSL) models that combines interpretability of SSMs and flexibility of LSTMs.

The key intuition, motivated by SSMs, is to learn dynamics in the state space, rather than in the sample space. However, we do not assume transition in the state space is linear, Gaussian, or Markovian. Existing approaches such as the extended Kalman filter (EKF) attempted to work with a general nonlinear transition function. Unfortunately, additional flexibility also introduced extra difficulty in the parameter estimation: EKF relies heavily on linearizing the nonlinear functions. We propose to use LSTM to model the dynamics in the latent state space, as they can learn from complex sequences without making limiting assumptions. The BPTT algorithm is also well established so that no additional approximation is needed in training the latent dynamics.

**Generative process**    Let $h(\cdot)$ be the emission function that maps a latent state to a parameter of the sample distribution. As illustrated in Figure 1 (a), the generative process of SSL for a single sequence is:

- For $t = 1, \ldots, T$:
    1. Perform LSTM transition: $s_t = \mathsf{LSTM}(s_{t-1}, z_{t-1})$
    2. Draw latent state: $z_t \sim p(z; g(s_t))$
    3. Draw observation: $x_t \sim p(x; h(z_t))$

The generative process specifies the following joint likelihood, with a similar factorization as SSMs except for the Markov transition:

$$p(x_{1:T}, z_{1:T}) = \prod_{t=1}^{T} p_\omega(z_t | z_{1:t-1}) p_\phi(x_t | z_t), \tag{5}$$

where $p_\omega(z_t | z_{1:t-1}) = p(z_t; g(s_t))$, $\omega$ is the set of parameters of $\mathsf{LSTM}(\cdot, \cdot)$ and $g(\cdot)$, and $\phi$ is the parameters of $h(\cdot)$. The structure of the likelihood function is better illustrated in Figure 1 (b), where each latent state $z_t$ is dependent to all previous states $z_{1:t-1}$ after substituting $s_t$ recursively. This allows the SSL to have non-Markovian state transition, with parsimonious parameterization thanks to the recurrent structure of LSTMs.

**Parameter estimation**    We continue with a single sequence for the ease of notation. A variational lower bound to the marginal data likelihood is given by

$$\log p(x_{1:T}) \geq \mathbb{E}_q \left[ \log \frac{p_\omega(z_{1:T}) p_\phi(x_{1:T} | z_{1:T})}{q(z_{1:T})} \right], \tag{6}$$

where $q(z_{1:T})$ is the variational distribution. Following the (stochastic) EM approach, iteratively maximizing the lower bound w.r.t. $q$ and the model parameters $(\omega, \phi)$ leads to the following updates:

- E-step: The optimal variational distribution is given by the posterior:

$$q^\star(z_{1:T}) \propto p_\omega(z_{1:T})p_\phi(x_{1:T}|z_{1:T}). \tag{7}$$

  In the case of LDS or HMM, efficient smoothing algorithms such as the RTS smoother or the forward-backward algorithm are available for computing the posterior expectations of sufficient statistics. However, without Markovian state transition, although the forward messages can still be computed, the backward recursion can no longer evaluated or efficiently approximated.

- S-step: Due to the difficulties in taking expectations, we take an alternative approach to collect posterior samples instead:

$$z^\star_{1:T} \sim q^\star(z_{1:T}), \tag{8}$$

  given only the filtering equations. We discuss the posterior sampling algorithm in detail in the next section.

- M-step: Given the posterior samples $z^\star_{1:T}$, which can be seen as Monte Carlo estimate of the expectations, the subproblem for $\omega$ and $\phi$ are

$$\omega^\star = \underset{\omega}{\arg\max} \, \log p_\omega(z^\star_{1:T}), \quad \phi^\star = \underset{\phi}{\arg\max} \sum_t \log p_\phi(x_t|z^\star_t), \tag{9}$$

  which is exactly the MLE of an LSTM, with $z^\star_{1:T}$ serving as the input sequence, and the MLE of the given emission model.

Having seen the generative model and the estimation algorithm, we can now discuss some instances of the proposed class of models. In particular, we provide two examples of SSL, for continuous and discrete latent states respectively.

**Example 1 (Gaussian SSL)** *Suppose $z_t$ and $x_t$ are real-valued vectors. A typical choice of the transition and emission is the Gaussian distribution:*

$$p(z_t; g(s_t)) = \mathcal{N}(z_t; g_\mu(s_t), g_\sigma(s_t)) \tag{10}$$

$$p(x_t; h(z_t)) = \mathcal{N}(x_t; h_\mu(z_t), h_\sigma(z_t)), \tag{11}$$

*where $g_\mu(\cdot)$ and $g_\sigma(\cdot)$ map to the mean and the covariance of the Gaussian respectively, and similarly $h_\mu(\cdot)$ and $h_\sigma(\cdot)$. For closed form estimates for the emission parameters, one can further assume*

$$h_\mu(z_t) = Cz_t + b, \quad h_\sigma(z_t) = R, \tag{12}$$

*where $C$ is a matrix that maps from state space to sample space, and $R$ is the covariance matrix with appropriate size. The MLE of $\phi = (C, b, R)$ is then given by the least squares fit.*

**Example 2 (Topical SSL, (Zaheer et al., 2017))** *Consider $x_{1:T}$ as the sequence of websites a user has visited. One might be tempted to model the user behavior using an LSTM, however due to the enormous size of the Internet, it is almost impossible to even compute a softmax output to get a discrete distribution over the websites. There are approximation methods for large vocabulary problems in RNN, such as the hierarchical softmax (Morin & Bengio, 2005). However, another interesting approach is to operate on a sequence with a "compressed" vocabulary, while learning how to perform such compression at the same time.*

*Let $z_t$ be the indicator of a "topic", which is a distribution over the vocabulary as in Blei et al. (2003). Accordingly, define*

$$p(z_t; g(s_t)) = \mathsf{Multinomial}(z_t; \mathsf{softmax}(Ws_t + b)) \tag{13}$$

$$p(x_t; h(z_t)) = \mathsf{Multinomial}(x_t; \phi_{z_t}), \tag{14}$$

*where $W$ is a matrix that maps LSTM states to latent states, $b$ is a bias term, and $\phi_{z_t}$ is a point in the probability simplex. If $z_t$ lies in a lower dimension than $x_t$, the LSTM is effectively trained over a sequence $z_{1:T}$ with a reduced vocabulary. On the other hand, the probabilistic mapping between $z_t$ and $x_t$ is interpretable, as it learns to group similar $x_t$'s together. The estimation of $\phi$ is typically performed under a Dirichlet prior, which then corresponds to the MAP estimate of the Dirichlet distribution (Zaheer et al., 2017).*

## 4    INFERENCE WITH PARTICLE GIBBS

In this section, we discuss how to draw samples from the posterior (7), corresponding to the S-step of the stochastic EM algorithm:

$$z_{1:T}^{\star} \sim p(z_{1:T}|x_{1:T}) = \frac{\prod_t p_\omega(z_t|z_{1:t-1})p_\phi(x_t|z_t)}{\int \prod_t p_\omega(z_t|z_{1:t-1})p_\phi(x_t|z_t)\,\mathrm{d}z_{1:T}}. \tag{15}$$

Assuming the integration and normalization can be performed efficiently, the following quantities can be evaluated in the forward pass without Markov state transition:

$$\alpha_t \equiv p(x_t|z_{1:t-1}) \propto \int p_\omega(z_t|z_{1:t-1})p_\phi(x_t|z_t)\,\mathrm{d}z_t \tag{16}$$

$$\gamma_t \equiv p(z_t|z_{1:t-1},x_t) \propto p_\omega(z_t|z_{1:t-1})p_\phi(x_t|z_t). \tag{17}$$

The task is to draw from the joint posterior of $z_{1:T}$ only given access to these *forward messages*.

One way to circumvent the tight dependencies in $z_{1:T}$ is to make a factorization assumption, as in Zaheer et al. (2017). More concretely, the joint distribution is decomposed as

$$\text{(factorization assumption)} \qquad p(z_{1:T}|x_{1:T}) \propto \prod_t p_\omega(z_t|z_{1:t-1}^{\mathrm{prev}})p_\phi(x_t|z_t), \tag{18}$$

where $z_{1:t-1}^{\mathrm{prev}}$ is the assignments from the previous inference step. However, as we confirm in the experiments, this assumption can be restrictive since the flexibility of LSTM state transitions is offset by considering each time step independently.

In this work, we propose to use a method based on SMC, which is a principled way of sampling from a sequence of distributions. More importantly, it does not require the model to be Markovian (Frigola et al., 2013; Lindsten et al., 2014). As described earlier, the idea is to approximate the posterior (15) with point masses, *i.e.*, weighted particles. Let $f(z_t|z_{1:t-1},x_t)$ be the importance density, and $P$ be the number of particles. We then can run Algorithm 1 with $\pi_t(z_{1:t}) = p(x_{1:t}, z_{1:t})$ being the unnormalized target distribution at time $t$, where the weight becomes

$$\alpha_t^p \propto \frac{p(z_{1:t}^p, x_{1:t})}{p(z_{1:t-1}^{a_{t-1}^p}, x_{1:t-1})f(z_t^p|z_{1:t-1}^{a_{t-1}^p},x_t)} = \frac{p_\omega(z_t^p|z_{1:t-1}^{a_{t-1}^p})p_\phi(x_t|z_t^p)}{f(z_t^p|z_{1:t-1}^{a_{t-1}^p},x_t)}. \tag{19}$$

As for the choice of the proposal distribution $f(\cdot)$, one could use the transition density $p_\omega(z_t|z_{1:t-1})$, in which case the algorithm is also referred to as the bootstrap particle filter. An alternative is the predictive distribution, a locally optimal proposal in terms of variance (Andrieu et al., 2010):

$$f^{\star}(z_t|z_{1:t-1},x_t) = \frac{p_\omega(z_t|z_{1:t-1})p_\phi(x_t|z_t)}{\int p_\omega(z_t|z_{1:t-1})p_\phi(x_t|z_t)\,\mathrm{d}z_t}, \tag{20}$$

which is precisely one of the available forward messages:

$$\gamma_t^p = f^{\star}(z_t|z_{1:t-1}^{a_{t-1}^p},x_t). \tag{21}$$

Notice the similarity between terms in (19) and (20). Indeed, with the choice of predictive distribution as the proposal density, the importance weight simplifies to

$$\alpha_t^p \propto \tilde{\alpha}_t^p = \int p_\omega(z_t|z_{1:t-1}^{a_{t-1}^p})p_\phi(x_t|z_t)\,\mathrm{d}z_t,, \tag{22}$$

which is not a coincidence that the name collides with the message $\alpha_t$. Interestingly, this quantity no longer depends on the current particle $z_t^p$. Instead, it marginalizes over all possible particle assignments of the current time step. This is beneficial computationally since the intermediate terms from (20) can be reused in (22). Also note that the optimal proposal relies on the fact that the normalization in (20) can be performed efficiently, otherwise the bootstrap proposal should be used.

After a full pass over the sequence, the algorithm produces Monte Carlo approximation of the posterior and the marginal likelihood:

$$\hat{p}(z_{1:T}|x_{1:T}) = \sum_p \alpha_T^p \delta_{z_{1:T}^p}(z_{1:T}), \quad \hat{p}(x_{1:T}) = \prod_t \frac{1}{P}\sum_p \tilde{\alpha}_t^p. \tag{23}$$

---

**Algorithm 2** Inference with Particle Gibbs

---

1. Let $z_0^p = z_0$ and $\alpha_0^p = 1/P$ for $p = 1, \ldots, P$.
2. For $t = 1, \ldots, T$:
   (a) Fix reference path: set $a_{t-1}^1 = 1$ and $z_{1:t}^1 = z_{1:t}^\star$ from previous iteration.
   (b) Sample ancestors $a_{t-1}^p \sim \alpha_{t-1}$ for $p = 2, \ldots, P$.
   (c) Sample particles $z_t^p \sim \gamma_t^p$ and set $z_{1:t}^p = (z_{1:t-1}^{a_{t-1}^p}, z_t^p)$ for $p = 2, \ldots, P$.
   (d) Compute normalized weights $\alpha_t^p$ for $p = 1, \ldots, P$.
3. Sample $r \sim \alpha_T$, return the particle path $z_{1:T}^{a_T^r}$.

---

The inference is completed by a final draw from the approximate posterior,

$$z_{1:T}^\star \sim \hat{p}(z_{1:T}|x_{1:T}), \tag{24}$$

which is essentially sampling a particle path indexed by the last particle. Specifically, the last particle $z_T^p$ is chosen according to the final weights $\alpha_T$, and then earlier particles can be obtained by tracing backwards to the beginning of the sequence according to the ancestry indicators $a_t^p$ at each position.

Since SMC produces a Monte Carlo estimate, as the number of particles $P \to \infty$ the approximate posterior (23) is guaranteed to converge to the true posterior for a fixed sequence. However, as the length of the sequence $T$ increases, the number of particles needed to provide a good approximation grows exponentially. This is the well-known depletion problem of SMC (Andrieu et al., 2010).

One elegant way to avoid simulating enormous number of particles is to marry the idea of MCMC with SMC (Andrieu et al., 2010). The idea of such Particle MCMC (PMCMC) methods is to treat the particle estimate $\hat{p}(\cdot)$ as a proposal, and design a Markov kernel that leaves the target distribution invariant. Since the invariance is ensured by the MCMC, it does not demand SMC to provide an accurate approximation to the true distribution, but only to give samples that are approximately distributed according to the target. As a result, for any fixed $P > 0$ the PMCMC methods ensure the target distribution is invariant.

We choose the Gibbs kernel that requires minimal modification from the basic SMC. The resulting algorithm is Particle Gibbs (PG), which is a conditional SMC update in a sense that a reference path $z_{1:T}^{\text{ref}}$ with its ancestral lineage is fixed throughout the particle propagation of SMC. It can be shown that this simple modification to SMC produces a transition kernel that is not only invariant, but also ergodic under mild assumptions. In practice, we use the assignments from previous step as the reference path. The final algorithm is summarized in Algorithm 2. Combined with the stochastic EM outer iteration, the final algorithm is an instance of the particle SAEM (Lindsten, 2013; Schön et al., 2015), under non-Markovian state transition.

We conclude this section by deriving forward messages for the previous examples.

**Example 1 (Gaussian SSL, continued)** *The integration and normalization preserves normality thanks to the Gaussian identify. The messages are given by*

$$\alpha_t = \mathcal{N}\left(x_t; Cg_\mu(s_t) + b, R + C[g_\sigma(s_t)]^{-1}C^T\right) \tag{25}$$

$$\gamma_t = \mathcal{N}\left(z_t; V\left(C^T R^{-1}(x_t - b) + [g_\sigma(s_t)]^{-1}g_\mu(s_t)\right), V\right), \tag{26}$$

*where $V = \left([g_\sigma(s_t)]^{-1} + C^T R^{-1} C\right)^{-1}$.*

**Example 2 (Topical SSL, continued)** *Let $\theta_t = \mathsf{softmax}(Ws_t + b)$. Since the distributions are discrete, we have*

$$\alpha_t \propto \langle \theta_t, \phi_{x_t} \rangle, \quad \gamma_t \propto \theta_t \circ \phi_{x_t}, \tag{27}$$

*where $\circ$ denotes element-wise product. Note that the integration for $\alpha_t$ corresponds to a summation in the state space. It is then normalized across $P$ particles to form a weight distribution. For $\gamma_t$ the normalization is performed in the state space as well, hence the computation of the messages are manageable.*

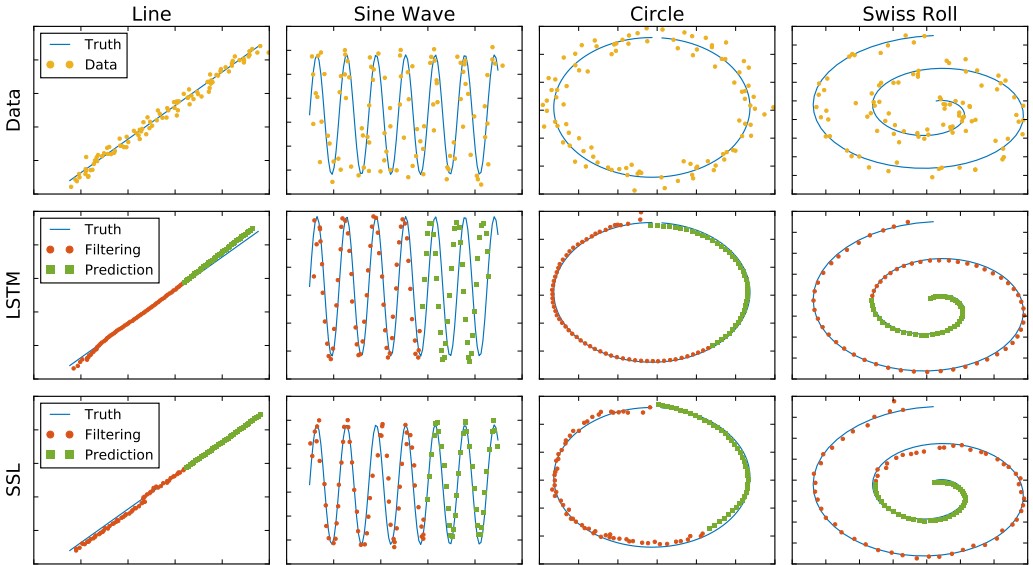

Figure 2: Tracking a synthetic trajectory. Top row: true trajectory and noisy observations. Middle row: training/testing performance of LSTM. Bottom row: training/testing performance of SSL.

## 5 EXPERIMENTS

We now present empirical studies for our proposed model and inference (denoted as SMC) in order to establish that (1) SSL is flexible in capturing underlying nonlinear dynamics, (2) our inference is accurate yet easily applicable to complicated models, and (3) it opens new avenues for interpretable yet nonlinear and non-Markovian sequence models, previously unthinkable. To illustrate these claims, we evaluate on (1) synthetic sequence tracking of varying difficulties, (2) language modeling, and (3) user modeling utilizing complicated models for capturing the intricate dynamics. For SMC inference, we gradually increase the number of particles $P$ from 1 to $K$ during training.

**Software & hardware** All the algorithms are implemented on TensorFlow (Abadi et al., 2016). We run our experiments on a commodity machine with Intel® Xeon® CPU E5-2630 v4 CPU, 256GB RAM, and 4 NVidia® Titan X (Pascal) GPU.

### 5.1 SYNTHETIC SEQUENCE TRACKING

To test the flexibility of SSL, we begin with inference using synthetic data. We consider four different dynamics in 2D space: (i) a straight line, (ii) a sine wave, (iii) a circle, and (iv) a swiss role. Note that we do not add additional states such as velocity, keeping the dynamics nonlinear except for the first case. Data points are generated by adding zero mean Gaussian noise to the true underlying dynamics. The true dynamics and the noisy observations are plotted in the top row of Figure 2. The first 60% of the sequence is used for training and the rest is left for testing.

The middle and bottom row of Figure 2 show the result of SSL and vanilla LSTM trained for same number of iterations until both are sufficiently converged. The red points refer to the prediction of $z_t$ after observing $x_{1:t}$, and the green points are blind predictions without observing any data. We can observe that while both methods are capturing the dynamics well in general, the predictions of LSTM tend to be more sensitive to initial predictions. In contrast, even when the initial predictions are not incorrect, SSL can recover in the end by remaining on the latent dynamic.

### 5.2 LANGUAGE MODELING

For Topical SSL, we compare our SMC inference method with the factored old algorithm (Zaheer et al., 2017) on the publicly available Wikipedia dataset, where documents with less than 500 words are excluded and the most frequent 200k word types are retained. We train on the first 60% of the

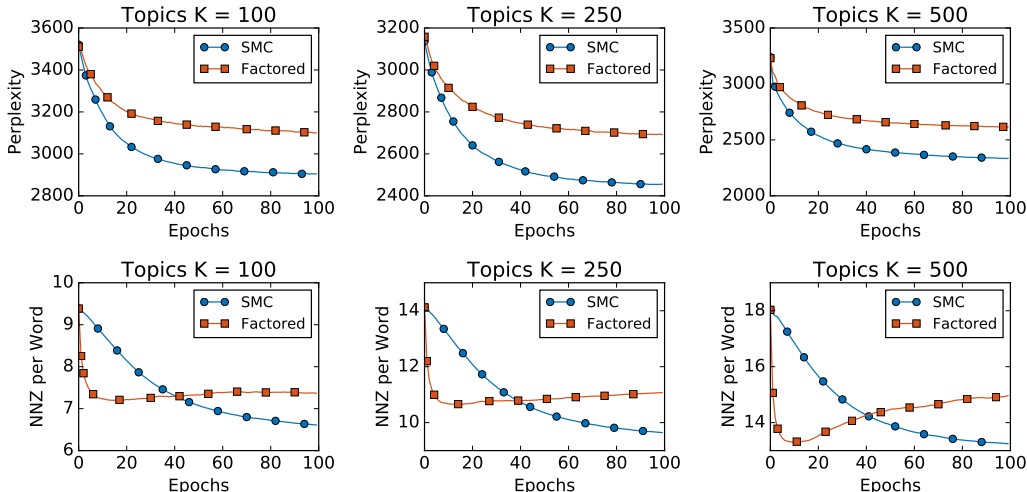

Figure 3: Comparison of the new inference method based on SMC to the older one assuming factored model. The top row represents perplexity on the held out set and the lower row represents the non zero entries in the word-topic count matrix. Lower perplexity indicates a better fit to the data and lower NNZ results in a sparser model and usually having better generalization.

documents and test on the rest, using the same settings in Zaheer et al. (2017). Figure 3 shows the test perplexity (lower is better) and number of nonzeros in the learned word topic count matrix (lower is better). In all cases, the SMC inference method consistently outperforms the old factored method. For comparison, we also run LSTM with the same number of parameters, which gives the lowest test perplexity of 1942.26. However, we note that LSTM needs to perform expensive linear transformation for both embedding and softmax at every step, which depends linearly on the vocabulary size $V$. In contrast, SSL only depends linearly on number of topics $K \ll V$.

**Ablation study** We also want to explore the benefit of the newer inference as dataset size increases. We observe that in case of natural languages which are highly structured the gap between factored approximation and accurate SMC keeps reducing as dataset size increases. But as we will see in case of user modeling when the dataset is less structured, the factored assumption leads to poorer performance. Also when the data size is fixed and the number of topics are varying, the SMC algorithm gives better perplexity compared to the old algorithm. Therefore we the SMC inference is consistently better in various settings.

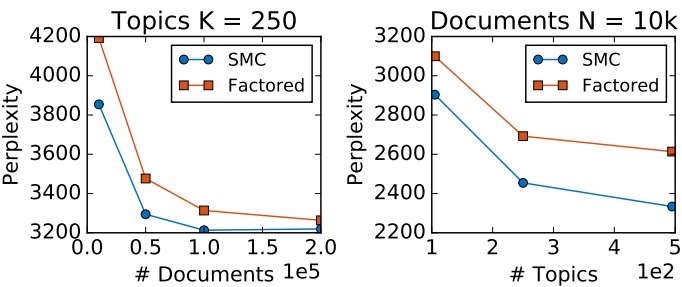

Figure 4: Comparison between SMC and factored algorithm by varying number of topics and documents

**Visualizing particle paths** In Figure 5, we show the particle paths on a snippet of an article about a music album [1]. As we can see from the top row, which plots the particle paths at the initial iteration, the model proposed a number of candidate topic sequences since it is uncertain about the latent semantics yet. However, after 100 epochs, as we can see from the bottom row, the model is much more confident about the underlying topical transition. Moreover, by inspecting the learned parameters $\phi$ of the probabilistic emission, we can see that the topics are highly concentrated on topics related to music and time. This confirms our claim about flexible sequence modeling while retaining interpretability.

---

[1] https://en.wikipedia.org/wiki/The_Haunted_Man_(album)

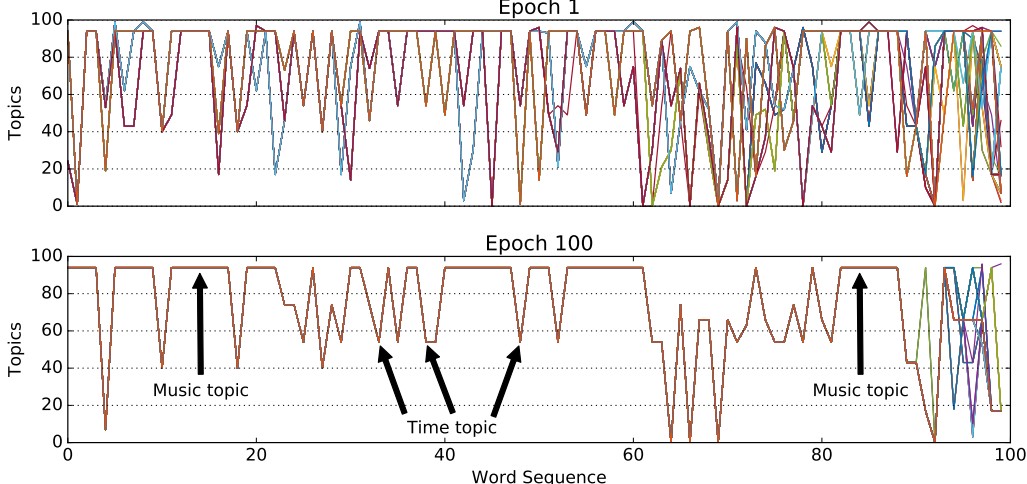

Figure 5: Particle paths of a document about a music album. Top row: at epoch 1. Bottom row: at epoch 100. After epoch 100 the document has converged to a sparse set of relevant topics.

## 5.3 USER MODELING

We use an anonymized sample of user search click history to measure the accuracy of different models on predicting users future clicks. An accurate model would enable better user experience by presenting the user with relevant content. The dataset is anonymized by removing all items appearing less than a given threshold, this results in a dataset with 100K vocabulary and we vary the number of users from 500K to 1M. We fix the number of topics at 500 for all user experiments. We used the same setup to the one used in the experiments over the Wikipedia dataset for parameters. The dataset is less structured than the language modeling task since users click patterns are less predictable than the sequence of words which follow definite syntactic rules. As shown in table 1, the benefit of new inference method is highlighted as it yields much lower perplexity than the factored model.

| Algorithm | # Users | |
|---|---|---|
| | 500k | 1M |
| Factored | 2430 | 2254 |
| SMC | 1464 | 1447 |

Table 1: Comparison on user data

## 6 DISCUSSIONS & CONCLUSION

In this paper we revisited the problem of posterior inference in Latent LSTM models as introduced in Zaheer et al. (2017). We generalized their model to accommodate a wide variety of state space models and most importantly we provided a more principled Sequential Monte-Carlo (SMC) algorithm for posterior inference. Although the newly proposed inference method can be slower, we showed over a variety of dataset that the new SMC based algorithm is far superior and more stable. While computation of the new SMC algorithm scales linearly with the number of particles, this can be naively parallelized. In the future we plan to extend our work to incorporate a wider class of dynamically changing structured objects such as time-evolving graphs.

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
