# OpenReview forum: "State Space LSTM Models with Particle MCMC Inference"
_ICLR.cc/2018/Conference — Reject_

### Official Review · AnonReviewer2 · 2017-11-26
**Poor model manipulation and missing many important references.**

**Rating:** 3
**Confidence:** 5

**Review:**

This article presents an approach for learning and inference in nonlinear state-space models (SSM) based on LSTMs. Learning is done using a stochastic EM where Particle PMCM is used to sample state trajectories.

The model is presented assuming that SSMs are linear. This is not necessarily the case since nonlinear SSMs have been used for a long time (see for example Ljung, 1999, "System Identification, Theory for the User"). The presented model is a nonlinear SSM with a particular structure that uses LSTMs.

The model described in the paper is Markovian: if one defines the variable sz_t = {s_t, z_t} there exists a Markov chain for the latent state sz:

sz_t -> sz_{t+1} -> sz_{t+2} -> ...

Marginalizing the latent variables s_t leads to a structure that, in general, is not Markovian. The authors claim that this marginalization "allows the SSL to have non-Markovian state transition". The word "allows" may mislead the reader in thinking that the model has gained some appealing property whereas the model is still essentially Markovian as evidenced by the Markov chain in sz. Any general algorithm for inference in nonlinear Markovian models could be used for inference of sz.

The algorithm used for inference and learning is stochastic EM with PMCMC but the authors do not cite important prior work such as: Lindsten (2013) "An efficient stochastic approximation EM algorithm using conditional particle filters"


Pros:

The model is sound.

The overall structure of the paper is good.


Cons:

The authors formulate the problem in such a way that they are forced to use an algorithm for non-Markovian models when they could have conserved the Markovian structure by choosing the appropriate parameterization.

The presentation of state-space models, filtering and smoothing shows some lack of familiarity with the literature. The control theory literature has dealt with nonlinear SSMs for decades and there is recent work in the machine learning community on nonlinear SSMs, e.g. Gaussian Process SSMs.

I would advise against the use of non-English expressions unless they are used precisely:

   - sine qua non: LSTMs are not literally an indispensable model for sequence modeling nowadays. If the use of Latin was unavoidable, "de facto standard" would have been slightly more accurate.

   - bona fide: I am not sure what the authors wanted to say.

   - naívely: the correct spelling would be naïvely or naively.

---

> ### Author Response · Authors · 2017-12-25
> **Some clarifications**
>
> We thank the reviewer for detailed comments, particularly the important references and the usage of latin phrases. However, we would also like to clarify some misunderstandings regarding the paper.
>
> In the paper, we proposed an instantiation of non-linear non-Markov state space model where the transition probabilities are defined using an LSTM. There are several prior work about non-linear SSM, like those referenced in first paragraph of Related Works and and the second paragraph of Section 3. For example we discussed EKF for general nonlinear transition/emission functions (Reviewer1 also points out limitation of EKF: only applicable to Gaussian noise).  However, such models did not cater to our need of being able to handle structured discrete data while at the same time have long history dependency. Therefore we are certainly not claiming that the proposed model is the first “nonlinear extension” to SSMs. Rather, we consider LSTM as yet another form of nonlinear transition function, but a particularly interesting one that shows outstanding performance in sequence modeling.
>
> Furthermore, the LSTM transition function brings not only nonlinearity, but also non-Markovianity, which is the next point we would like to clarify. Indeed, it is true that in the joint space of LSTM state and SSM state, the model is Markov. However, it does not do justice to say such jointly Markov model does not bring in any appealing property. Consider LSTM: in the joint space of LSTM state and observation, the model is Markov as well, since conditioned on the pair (state, observation) at time t, the pair at time t-1 is independent of the pair at time t+1, but this view of LSTM gives no insight and no inference has been proposed using this view. One can compare this to a vanilla Markov chain over the observation space, for instance the bigram language model. The gain brought by the jointly Markov model (LSTM) over the marginally Markov model (bigram) is apparent.
>
> Last but not least, the question arises that since the proposed model is jointly Markov in (s_t, z_t), why not use algorithm that assumes Markovianity. Indeed, one could derive a particle smoothing algorithm for the pair (s_t, z_t), however it has O(P^2) time complexity, where P is the number of particles (Schön et al. 2015). Although there exist methods that reduce the time complexity of particle smoothers, such as (Klaas et al., 2006), they still rely on asymptotics over P. As noted in the paper, the choice of particle gibbs is not only to accommodate non-Markov transition, but also to avoid simulating too many particles.
>
> We appreciate the suggestions on the latin phrases. They are fixed in the updated draft.
>
>
> References:
> Schön, Thomas Bo, et al. "Sequential Monte Carlo Methods for System Identification." Proceedings of the 17th IFAC Symposium on System Identification, Beijing, China, October 19-21, 2015.. Vol. 48. 2015.
> Klaas, M., Briers, M., De Freitas, N., Doucet, A., Maskell, S., & Lang, D. "Fast particle smoothing: If I had a million particles." Proceedings of the 23rd international conference on Machine learning. ACM, 2006.

---

### Official Review · AnonReviewer1 · 2017-11-27
**PMCMC EM for LSTM-based SSM with unclear contributions**

**Rating:** 5
**Confidence:** 5

**Review:**

[After author feedback]
I would suggest that the authors revise the literature study and contributions to more accurately reflect prior work.

[Original review]
The authors propose state space models where the transition probabilities are defined using an LSTM. For inference the authors propose to make use of Monte Carlo expectation maximization.

The model proposed seems to be a special case of previously proposed models that are mentioned in the 2nd paragraph of the related works section, and e.g. the Maddison et al. (2017) paper. The inference method has also been studied previously (but not to my knowledge applied to SSLs/SRNNs), see the following review papers and references therein:
Schön, Lindsten, Dahlin, W˚agberg, Naesseth, Svensson, Dai, "Sequential Monte Carlo Methods for System Identification", 2015
Kantas, Doucet,  Singh, Maciejowski, Chopin, "On Particle Methods for Parameter Estimation in State-Space Models", 2015

Given this it is unclear to me what the novel contributions are. Perhaps the authors can elaborate on this?

Minor comments:
- Note that generally a state space model only has the Markov assumption, there is no restrictions on the transition and observation models.
- EKF also requires Gaussian noise
- It is a bit unclear what is meant by "forward messages" e.g. below eq. (6). For this model I believe the exact would generally be unavailable (at least for continuous models) because they would depend on previous messages.
- Eq. (12) and (14) are exactly the same? The text seems to indicate they are not.
- The optimal proposal is only locally optimal, minimizing the incremental weight variance
- "w" should be "x" in eq. (20)

---

> ### Author Response · Authors · 2017-12-25
> **Contributions of the Paper**
>
> We thank the reviewer for valuable comments and detailed feedback.
>
> We would like to highlight the main contributions of the work:
>
> - As correctly pointed out by the reviewer, we proposed a simple framework of state space models where the transition probabilities are defined using an LSTM and observation probabilities are parametric. Among other advantages, this design enables ease in handling structured discrete data and discrete latent variables, unlike plethora of existing work on stochastic RNNs and its variants. These latter models extend RNN by combining with a deep generative model such as VAE at the output layer, which allows for impressive performance on structured continuous data such as image and sound, handling structured discrete data, but handling discrete latent variables is not as straightforward as in SSM. (In fact, stochastic gradient estimator for discrete latent variables is an active research direction, for instance Gumbel-softmax/Concrete distribution, REBAR, RELAX estimators.)
>
> - In the proposed model as the transition probabilities are defined using an LSTM (as correctly pointed out by the reviewer), consequently the model is not Markovian. Thus, existing works for nonlinear SSMs, for instance in (Lindsten, 2013) and (Schön et al., 2015) assume a Markov transition in the derivation of the algorithm, which is not suitable for our proposed model. We show that even under non-Markov state transition, particle filter or particle gibbs can be used, and furthermore not only the bootstrap proposal, but also the locally optimal proposal can be efficiently evaluated in some examples.
>
> At a high level, we demonstrated a way to enhance a classical Bayesian model (good for interpretability and structured discrete data) with the flexibility of deep neural networks.
>
> Regarding the confusion on the “forward messages”, we will clarify them by clearly defining them to be the quantities that are computable in the forward pass, as in forward-backward message passing algorithm. As noted in Example 4.1, the messages are available in closed form for linear Gaussian case. Note that this does not necessarily mean restricted flexibility of the state transition, since the rich function class of LSTM is encoded in g(s). This is a similar to VAE in spirit, which also uses Gaussian as the variational distribution.
>
> We also thank the reviewer for pointing out the typos. In the equation for factorization assumption, the conditioned past z variables was meant to be the assignments from the previous iteration. Indeed, such factorization does not hold, which is why it is an “assumption”. This was fixed in the updated draft. Also we appreciate the reviewer for pointing out another plus point for the proposed work that EKF is limited to Gaussian noise, but no such limitation exists for SSL.
>
>
> References:
> Lindsten, Fredrik. "An efficient stochastic approximation EM algorithm using conditional particle filters." Acoustics, Speech and Signal Processing (ICASSP), 2013 IEEE International Conference on. IEEE, 2013.
> Schön, Thomas Bo, et al. "Sequential Monte Carlo Methods for System Identification." Proceedings of the 17th IFAC Symposium on System Identification, Beijing, China, October 19-21, 2015.. Vol. 48. 2015.

---

> > ### Comment · AnonReviewer1 · 2018-01-02
> > **Inference for non-Markovian models using particle filters and PMCMC**
> >
> > Thank you for your response. Note that while the papers mentioned focus on Markovian models there is nothing limiting their use to this specific class of models:
> >
> > For examples where PSAEM has been applied to non-Markovian models see e.g.
> > * Frigola et al., Identification of Gaussian Process State-Space Models with Particle Stochastic Approximation EM, 2014
> > * Svensson et al., Identification of jump Markov linear models using particle filters, 2014
> >
> > For examples where particle filters and PMCMC methods have been applied to non-Markovian models, see e.g.
> > * Wood et al., A new approach to probabilistic programming inference, 2014
> > * Naesseth et al., Sequential Monte Carlo for Graphical Models, 2014
> > * Lindsten et al., Particle Gibbs with Ancestor Sampling, 2014

---

> > > ### Author Response · Authors · 2018-01-03
> > > **Re:**
> > >
> > >
> > > We thank the reviewer for the comment and the point is taken.
> > >
> > > However, we would like to mention that we are aware of the papers referred to above. For example, in the paragraph above Eq. (19), (Frigola et al. 2013) and (Lindsten et al. 2014) are cited as examples for the application of particle methods in non-Markov models. To re-emphasize, by **no** means we are claiming that the proposed algorithm is the first application of particle methods for non-Markov models.
> > >
> > > We believe that matching inference procedures to models is not trivial and is an art. This is exemplified by the plethora of papers being published, c.f. (Lindsten & Schön, 2013) inter alia, including the ones pointed out by the knowledgeable reviewer, for applying particle filters and PMCMC methods to various different but aptly chosen models. Every small detail matters.
> > >
> > > As stated earlier, our goal is to enhance classical Bayesian model with the flexibility of deep neural networks, and apply the right inference algorithm, which is both rigorous and practical, and furthermore does not need strong assumptions (e.g. factorized conditionals, biased gradient approximation, etc). Moreover, to the best of our knowledge, the use of particle inference methods in neural sequence models (RNN/LSTM) is novel.
> > >
> > > We did not use particle smoothing because it requires O(P^2) complexity for P particles. Kindly refer to the second last paragraph of our response to AnonReviewer2.
> > >
> > > References:
> > > Lindsten, F., & Schön, T. B. (2013). Backward Simulation Methods for Monte Carlo Statistical Inference. Foundations and Trends in Machine Learning, 6(1), 1–143.

---

> > > > ### Comment · AnonReviewer1 · 2018-01-09
> > > > **Re:**
> > > >
> > > > Note that the Frigola et al. (2013) does approximate Bayesian inference using PGAS, whereas the 2014 paper I mentioned does it using PSAEM which is highly related to the way you propose to do inference.
> > > >
> > > > This is not the first paper that proposes particle inference for LSTM/RNN-based models, see e.g. FIVO/AESMC/VSMC papers as well as the Gu et al. "Neural Adaptive Sequential Monte Carlo".
> > > >
> > > > Note further that many of the methods for particle filter-based learning described in the two references in the original review can be applied to the model without a P^2 complexity. The model proposed can (as R2 pointed out) be interpreted as Markovian with a degenerate transition distribution. It is well-known that SMC-based methods can be straightforwardly applied in this case.

---

### Official Review · AnonReviewer3 · 2017-11-30
**A nice formulation of a stochastic LSTM**

**Rating:** 7
**Confidence:** 5

**Review:**

This paper introduces a novel extension of the LSTM which incorporates stochastic inputs at each timestep. These stochastic inputs are themselves dependent on the LSTM state at the previous timestep. Considering the stochastic dependencies, this then yields a highly flexible non-Markov state space model, where the latent variable transitions are partially parameterized by an LSTM update.

Naturally, the challenges are then efficiently estimating parameters and performing inference over the latent states. Here, SMC (and conditional SMC / particle Gibbs) are used for inference over the latent states z. A particularly nice touch is that even when the LSTM model is used for the transitions in the latent space, so long as the conditional distributions p(z_t | z_{1:t-1}) are conjugate with the emission distribution then it is possible to compute the optimal forward filtering proposal distribution in closed form, as done for the conditionally Gaussian (with affine Gaussian observations) and conditionally multinomial models considered here. Note that this really is a special feature of the models under consideration, though: for example, if the emission distribution p(x_t | z_t) is instead a *nonlinear* Gaussian, then one would have to fall back to bootstrap proposals. This probably deserves some mention: equations (13) are not, generally, tractable to integrate or normalize.

I think this paper is missing a few necessary details on how the overall optimization algorithm proceeds, which I would like to see in an update. I understand that particle Gibbs updates (or SMC) are used to approximate the posterior distribution in a Monte Carlo EM algorithm. However, this does leave some questions:

1. For the M step, how are the \omega parameters (of the LSTM) handled in equation (8)? I understand that due to the particular models considered, maximum likelihood estimates of \phi can be found in closed form. However, that’s not the case for \omega. Is a gradient descent algorithm run to convergence? Or is a single gradient step taken, interleaved with a single PG update? Or something else?

2. How reliably does the algorithm as a whole converge? Monte Carlo EM does not in general have convergence guarantees of “standard” EM (i.e. each step is not guaranteed to monotonically improve the lower bound). This might be fine! But, I think requires a bit of discussion.

3. Is it necessary to include a replenishing operation (or independent MCMC steps) in the particle Gibbs algorithm? A known issue when running an iterated conditional SMC algorithm like this is that path degeneracy can make it very difficult for the PG kernel to mix well over the early time steps in the LSTM. Does this issue appear here? How many particles P are needed to efficiently mix, when considering time series of length T?

---

> ### Author Response · Authors · 2017-12-25
> **Clarifications on Inference**
>
> We thank the reviewer for the insightful comments and raising important questions. We are glad that reviewer found the work to be novel and having a “nice touch”. Kindly find below response to the question.
>
> The inference procedure presented in the paper is not an ad-hoc method. We would be happy to provide more discussion about this in the paper. Our overall inference scheme is an instantiation of stochastic generalized EM (Neal et al, 1998). Such methods have been theoretically studied in detailed c.f. (Nielsen, 2000), (Delvon et al. 1999). We agree with the reviewer that such methods do not possess the property of monotonically increasing the lower bound, however under certain regularity conditions (which are met if we have LSTM and exponential family observation) these method in expectation reach a critical point. With more assumptions, even stronger results have been proved.
>
> Further in the M step one need not find the optimizer but just improve the likelihood in expectation. This can be achieved, e.g., by taking a few number of stochastic gradient steps, as we did for LSTM updates. To be specific in case of application to discrete data (Example 2), we made a pass over the dataset  whereas for phi we used the closed form optimizer (Note the optimization for LSTM and phi are independent given z).
>
> Initially we also suspected some kind of path degeneracy to occur. However in our experiments, we did not see the need for a replenishing operation. In particular, we started off with a variant of PG called PGAS (particle gibbs ancestral sampling) by (Lindsten et al. 2014), which specifically targets to resolve the path degeneracy issue in PG. We tried the approximation for non-Markovian model as mentioned in (Lindsten et al. 2014) with lag = 1, however it did not provide significant improvement over much faster and simple strategy of increasing the number of particles P from 1 to K during training.  In general we observed that in the initial phase the particles do not collapse towards a single path; however after 100 epochs the proposed particle paths agree at most of the time points (Please refer to Figure 5 for an illustration).
>
> Also we will fix small typos and add clarifications regarding the non-conjugate cases when the marginalization in alpha message cannot be computed in closed form and the normalization cannot be performed efficiently, that one would have to resort to methods like bootstrap proposals.
>
>
> References:
> Neal, Radford M., and Geoffrey E. Hinton. "A view of the EM algorithm that justifies incremental, sparse, and other variants." Learning in graphical models. Springer Netherlands, 1998. 355-368.
> Nielsen, Søren Feodor. "The stochastic EM algorithm: estimation and asymptotic results." Bernoulli 6.3 (2000): 457-489.
> Delyon, Bernard, Marc Lavielle, and Eric Moulines. "Convergence of a stochastic approximation version of the EM algorithm." Annals of statistics (1999): 94-128.
> Lindsten, Fredrik, Michael I. Jordan, and Thomas B. Schön. "Particle gibbs with ancestor sampling." Journal of Machine Learning Research 15.1 (2014): 2145-2184.

---

### Decision · Program_Chairs · 2018-01-29
**ICLR 2018 Conference Acceptance Decision**

**Decision:**

Reject

**Comment:**

Thank you for submitting you paper to ICLR. The consensus from the reviewers is that this is not quite ready for publication. The work is related to (although different from) Gu et al Neural Sequential Monte Carlo NIPS2015 and it would be useful to point this out in the related work section.